# Photosynthetic Behavior of *Argania spinosa* (L.) Skeels Induced under Grazed and Ungrazed Conditions

Aicha Nait Douch [1], Laila Boukhalef [1], Abdelhafed El Asbahani [2], Ali A. Al-Namazi [3],
Khadija El Mehrach [4], Laila Bouqbis [1], Mourad Touaf [1] and Fatima Ain-Lhout [1,*]

1 Equipe d'Ecologie et Sciences de l'Environnement, Faculté des Sciences Appliquées, Université Ibn Zohr, Azrou 86153, Morocco
2 Laboratoire de Chimie Appliquée et Environnement, Faculté des Sciences, Université Ibn Zohr, Agadir 80000, Morocco
3 King Abdulaziz City for Science and Technology (KACST), P.O. Box 6086, Riyadh 11442, Saudi Arabia
4 Laboratoire de Biotechnologies Végétales, Faculté des Sciences, Université Ibn Zohr, Agadir 80000, Morocco
* Correspondence: fzainlhout@gmail.com; Tel.: +212-6660-21070

**Abstract:** The endemic Moroccan species *Argania spinosa* is considered the most grazed tree species in its distribution area. Since grazing exerts an important effect on plant performances, we attempted to explore the impact of grazing on *A. spinosa*. Thus, we performed a comparative field experiment where seasonal variations of gas exchange, photochemical efficiency, relative water content, photosynthetic pigment content, and stomatal features were assessed in grazed and ungrazed trees. The net photosynthetic rate was increased in grazed trees in spring and autumn, the favorable seasons. Enhancement of photosynthetic performance may be due to the high stomatal conductance registered in grazed trees. This mechanism may compensate for the lost leaf area, in order to recover from grazing stress. In addition, grazed trees exhibit a better photochemical efficiency, use water more economically and show lower oxidative stress. However, results obtained in summer show that the compensation mechanism could be limited by summer drought. Since the key to preserving the future of forests is sustainable forest management, our results suggest that proper grazing management can be a control tool to increase plant performance and improve species resilience.

**Keywords:** net photosynthesis; stomatal conductance; water use efficiency; photochemical efficiency; chlorophyll content

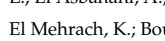



## 1. Introduction

*Argania spinosa* (L.) Skeels is the only representative endemic species of the tropical family of "Sapotaceae" in Morocco. The Argan forests are the second forestry stands in the country, as they extend into arid and semi-arid areas in the west central area of Morocco [1]. *A. spinosa* is well adapted to arid environments, exhibiting elevated productivity even in the summer season [2,3]. This species plays a crucial role, especially for the local population, in terms of its ecological, botanical, economical, and social interests [4–7]. Argania is a multi-purpose benefits tree, because each part (wood, leaves, fruits, oil) is used as a product or by-product, thus representing an important source of income, food, and even animal feed for the local population [5]. It was designated a UNESCO Biosphere Reserve in December 1998. The 10 May was declared the international day of *A. spinosa* by the United Nations General Assembly in 2021.

Climate change, urbanization, overuse of resources, and extension of agriculture and rangelands are currently leading to the degradation of most arid and semi-arid forests [8]. Indeed, despite the importance of the Argan tree, approximately half of the Argan forest area disappeared during the 20th century. The regression of Argan forests is mainly due to desertification, pastoral activities, and the overexploitation of forest resources by the local

population [8–10]. Argan tree contribution to the diet of goats ranges from 47% to 84%, depending on the climatic conditions and time of year [11].

Livestock grazing is an important biological factor determining the structure and composition of ecosystems [12]. Grazing generally has a negative effect on plants [13–17].In a previous work conducted in Admine forest, a heavily humanized area traditionally managed for livestock raising and agriculture. Ain-Lhout et al. [18] found that high intensity of herbivory pressure and human use of Argan trees reduced tree size and crow cover of *A. spinosa.* In another work on the same species and the same environmental context. Zunzunegui et al. [19] found that high herbivory pressure induces a pronounced decrease in fruit production. Indeed, herbivores cause structural damage, remove biomass of plants, decrease the leaf surface available for photosynthesis, and have led to excessive degradation of semi-arid and arid grasslands worldwide [20].

Nonetheless, several studies have shown that grazing pressure may have a positive effect on the performance of plants, especially on the primary production [21–26]. This positive effect is called grazing optimization [22], which represents the plant response to continuous grazing pressure [23]. In several case studies, compensatory photosynthesis has been described. This phenomenon is defined as a rise in foliage photosynthetic rates on damaged plants relative to the foliage of undamaged plants [25].

Grazing effect on ecosystems and vegetation communities is reflected by individual species responses; the dominant species have a crucial role [27]. At the individual level, injury by grazing may cause a large range of physiological and metabolic responses [28]. To defend themselves against herbivores, the plant can increase their fitness through physiological and morphological mechanisms, such as toxic chemicals, physical barriers to an animal attack, mobilization of reserve storage, or increased photosynthetic rate [29,30].

Studies on the ecological impact of overgrazing in arid and semi-arid regions are not convincing and often contradictory. Consequently, more information is needed on the effects of grazing on these dry forest ecosystems for a better understanding of their functioning and, therefore, to promote the management of sustainable livestock grazing.

Deepening knowledge on tree physiological adaptation to grazing is needed for sustainable Argan forest management. Thus, the main objective pursued in this study was to explore the effects of grazing on the photosynthetic performances of *A. spinosa* in an arid climate. For this purpose, gas exchange, fluorescence, photosynthetic pigment content, relative water content, and stomatal features were seasonally measured in grazed and ungrazed trees from the same area. We assumed that the Argan tree should have a differential photosynthetic response to face herbivory pressure. As a tolerant specie to herbivores [19], *A. spinosa* will exhibit compensatory photosynthesis under grazing conditions during the favorable seasons. However, we expect that the summer drought may limit the compensation mechanism.

## 2. Results

The annual mean temperature and rainfall recorded during the experiment were 22.55 °C and 51 mm, respectively (Figure 1). The hydrological year during the study period was extremely dry.

The two-way ANOVA analyses showed significant differences between grazed and ungrazed sites throughout the study period. Moreover, most parameters exhibited seasonal variation (Table 1).

$A_{Net}$ was significantly higher for grazed plants of *A. spinosa* in spring and autumn, with, respectively, $p < 0.001$ and $p < 0.05$ (Figure 2a). In summer, plants from both sites showed the lowest value of the year. Two-way ANOVA revealed that $A_{Net}$ proved significant differences between sites, seasons, and the interaction between site and season ($p < 0.001$). Note that the high standard deviation values are due to the high genetic diversity described in this species [31].

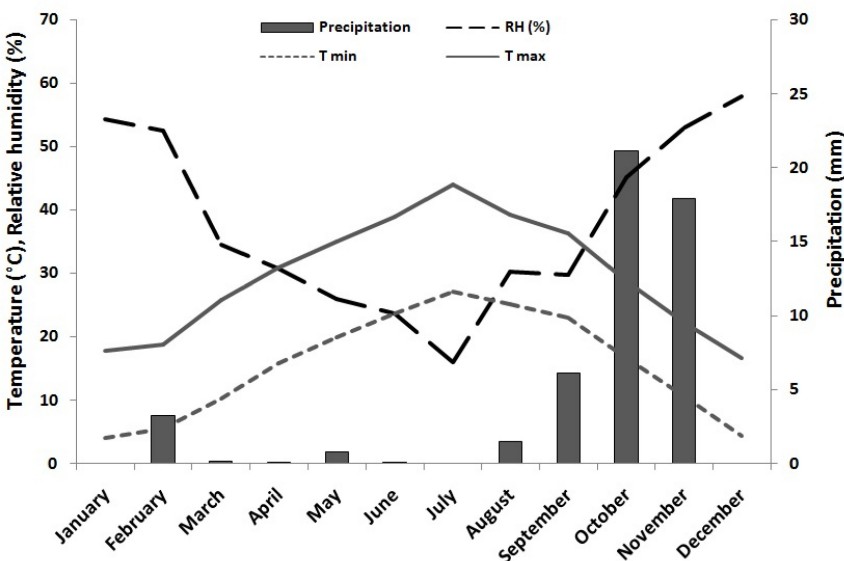

**Figure 1.** Daily maximum and minimum mean temperatures, air relative humidity, and monthly precipitation during the study period (2018) in M'sguina forest.

**Table 1.** Two-way ANOVA results on the effects of site, month, and the interaction between site and month for the physiological studied variables (* significant result).

| Variables | Site | | | Season | | | Site × Season | | |
|---|---|---|---|---|---|---|---|---|---|
| | fd | F | *p* | fd | F | *p* | fd | F | *p* |
| $A_{Net}$ | 1 | 31.08 | 0.001 * | 2 | 46.94 | 0.001 * | 2 | 8.48 | 0.001 * |
| $g_s$ | 1 | 8.39 | 0.05 * | 2 | 186.78 | 0.001 * | 2 | 49.67 | 0.001 * |
| $WUE_i$ | 1 | 10.31 | 0.05 * | 2 | 29.13 | 0.001 * | 2 | 4.17 | 0.05 * |
| $C_i$ | 1 | 13.38 | 0.001 * | 2 | 3.83 | 0.05 * | 2 | 2.35 | 0.098 |
| Fv/Fm | 1 | 24.37 | 0.001 * | 2 | 2130.83 | 0.001 * | 1 | 1.484 | 0.224 |
| ΦPS(II) | 1 | 21.23 | 0.001 * | 2 | 221.71 | 0.001 * | 1 | 16.350 | 0.001 * |
| Chla+b | 1 | 13.345 | 0.001 * | 2 | 188.755 | 0.001 * | 2 | 24.073 | 0.001 * |
| Chla/b | 1 | 1.029 | 0.312 | 2 | 7.4137 | 0.001 * | 2 | 1.508 | 0.224 |
| Car | 1 | 16.32 | 0.001 * | 2 | 24.22 | 0.001 * | 2 | 15.33 | 0.001 * |
| Chla+b/Car | 1 | 116.79 | 0.001 * | 2 | 79.29 | 0.001 * | 2 | 7.32 | 0.001 * |
| RWC | 1 | 0.06 | 0.106 | 2 | 25.98 | 0.001 * | 2 | 3.774 | 0.08 |
| SD | 1 | 21.622 | 0.001 * | 2 | 4.457 | 0.05 * | 2 | 8.004 | 0.001 * |

Stomatal conductance ($g_s$) showed a similar pattern to that of $A_{Net}$, except in spring, with a significantly high value recorded in autumn (Figure 2b), in the grazed population. Significant differences were recorded between the grazed and ungrazed sites in autumn ($p < 0.001$).

$C_i$ showed higher values in grazed plants during spring, summer, and autumn (Figure 2d). Furthermore, significant differences were detected between the two localities during spring ($p < 0.05$) and autumn ($p < 0.001$).

$WUE_i$ was significantly higher ($p < 0.05$) in grazed plants during spring and autumn, while the lowest $WUE_i$ values were recorded in summer at both sites (Figure 2c).

Maximal PSII quantum yield (Fv/Fm) exhibited values over 0.7 during spring and autumn in both sampling sites (Figure 3a). However, Fv/Fm values decreased to $0.63 \pm 0.09$ and $0.69 \pm 0.06$ in summer, respectively, in ungrazed and grazed trees. Significant differences were recorded between the two sampling sites during spring and summer ($p < 0.001$). Effective PSII quantum yield (ΦPSII) showed a seasonal pattern in both sites, with the highest values recorded during spring and the lowest during summer (Figure 3b).

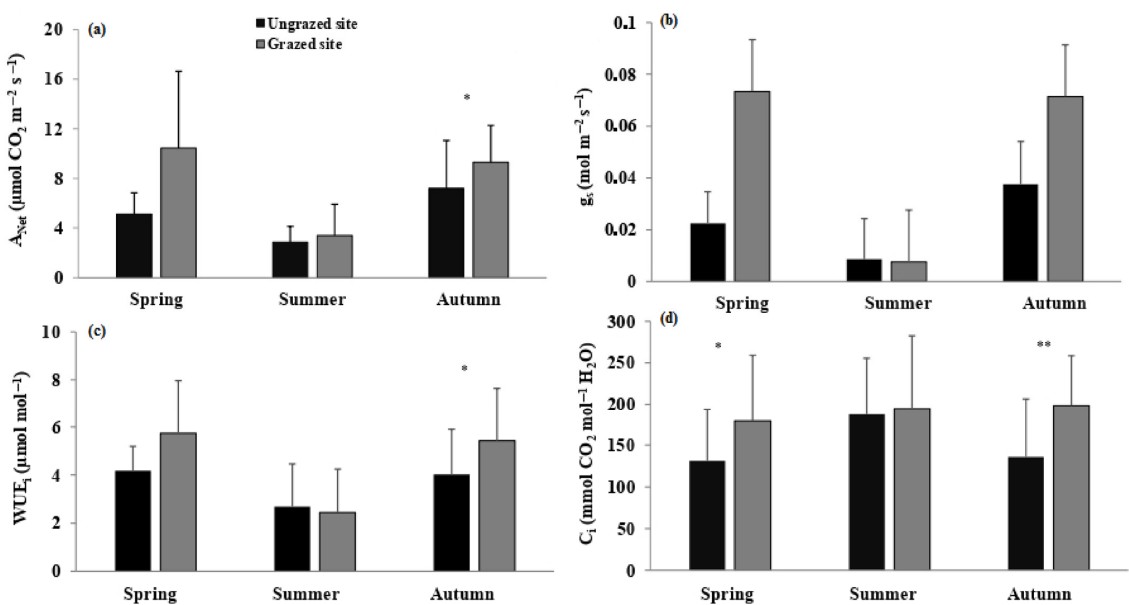

**Figure 2.** Mean (±SD) net photosynthetic rate ($A_{Net}$, (**a**)), stomatal conductance ($g_s$, (**b**)), instantaneous water use efficiency (WUEi, (**c**)), and sub-stomatal $CO_2$ concentration ($C_i$, (**d**)) for grazed and ungrazed Argan trees during the study period. The asterisks indicate significance levels comparing the two sites within each season by student's *t*-tests (** $p < 0.001$, * $p < 0.05$).

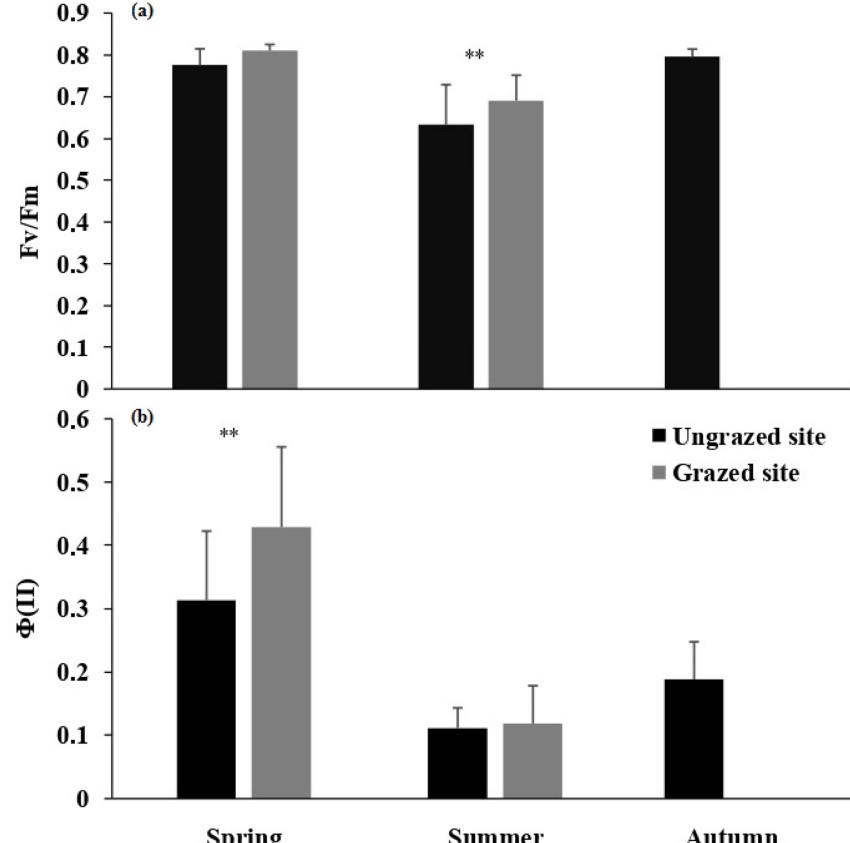

**Figure 3.** Mean (±SD) maximal PSII quantum yield (Fv/Fm, (**a**)), and effective PSII quantum yield (ΦPSII, (**b**)) for grazed and ungrazed Argan trees during the study period. The asterisks indicate significance levels comparing the two sites within each season by student's *t*-tests (** $p < 0.001$).

Total chlorophyll content (Chla+b) clearly showed a seasonal pattern in both sites (Figure 4a), with the highest values measured in autumn and the lowest in summer. In addition, grazed trees exhibited significantly higher chlorophyll content than ungrazed trees during spring ($p < 0.001$) and summer ($p < 0.05$). Chla/b showed the same pattern as total chlorophyll (Figure 4b), with a pronounced drop in ungrazed trees during summer. Significant differences were registered between sites in spring ($p < 0.05$) and summer ($p < 0.05$). Furthermore, the total chlorophyll/carotenoids ratio decreased in summer (Figure 4c). However, trees from the grazed site exhibited the highest values during all the studied periods, with significant differences between both sites ($p < 0.05$).

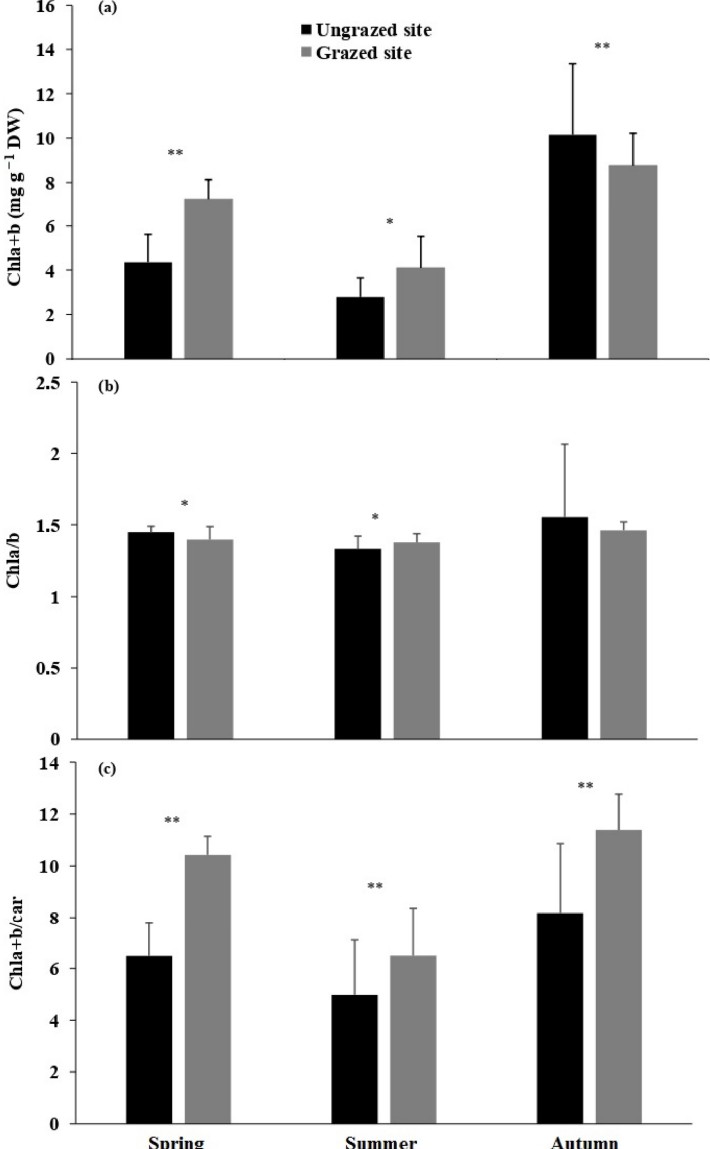

**Figure 4.** Mean (±SD) total chlorophyll content (Chla+b, (**a**)), chlorophyll a and b ratio, and (Chla/b, (**b**)) total chlorophyll and carotenoids ratio (Chla+b/car, (**c**)) for grazed and ungrazed Argan trees during the study period. The asterisks indicate significance levels comparing the two sites within each season by student's *t*-tests (** $p < 0.001$, * $p < 0.05$).

Relative water content (RWC) exhibited a seasonal pattern (Figure 5), with maximal values registered during autumn and minimum values during summer. No significant differences were detected between both sites.

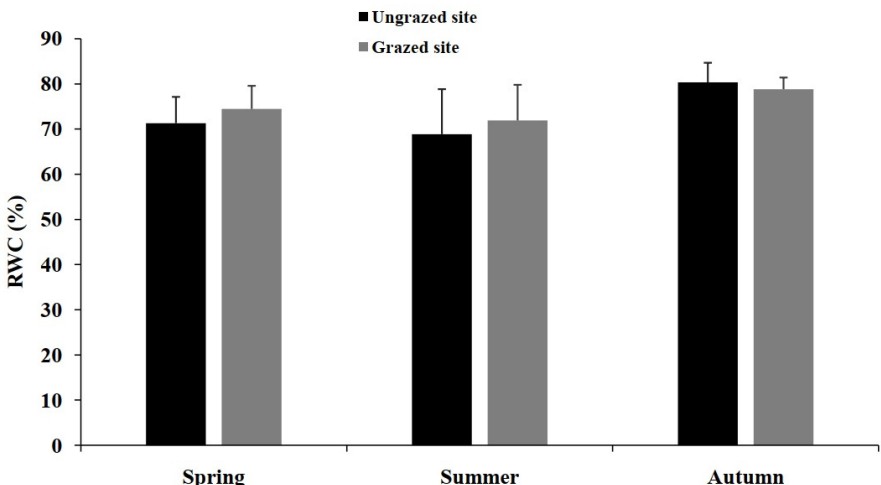

**Figure 5.** Mean (±SD) relative water content (RWC) for grazed and ungrazed Argan trees during the study period.

For stomatal density (SD), trees from the ungrazed site exhibited the highest spring values (Figure 6a). However, trees from the grazed site showed the lowest values during all the study periods. SD was significantly different in spring ($p < 0.001$) between both sites. The stomata length exhibited the highest significant values in grazed trees during spring season ($p < 0.05$) (Figure 6b).

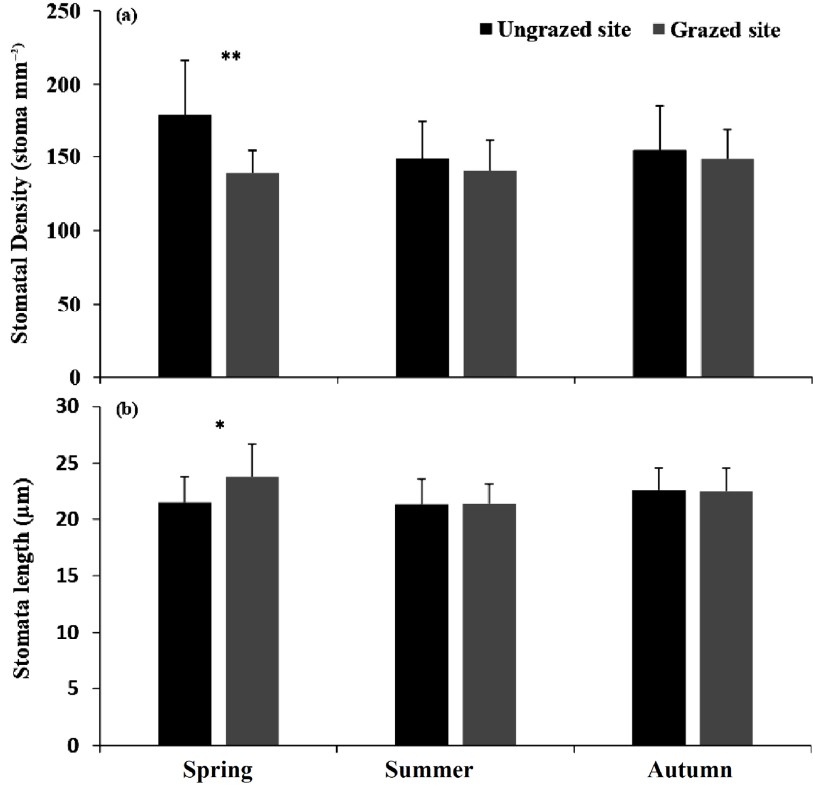

**Figure 6.** Mean (±SD) Stomatal density (SD, (**a**)) and stomata length (SL, (**b**)) for grazed and ungrazed Argan trees during the study period. The asterisks indicate significance levels comparing the two sites within each season by student's *t*-tests (** $p < 0.001$, * $p < 0.05$).

## 3. Discussion

Grazing is usually considered abiotic stress for many plant species as it can undermine plant fitness and growth status [32]. However, in our study, we found that grazing had a

significant effect on the photosynthetic performances of the Argan tree, and these effects changed through seasons. The two-way ANOVA test results showed significant differences between grazed and ungrazed trees in most of the parameters.

$A_{Net}$ values were significantly higher for grazed plants of *A. spinosa* during spring and autumn. The increased $g_s$ seems to provide an explanation for the simultaneous increased assimilation rate of photosynthesis. This enhancement of the plant photosynthetic activity following herbivore injury indicates that the leaves were functioning below their maximum photosynthetic potential and this is described as a tolerance mechanism and could compensate for the loss of leaf area [25]. Indeed, in a study under heavy grazing conditions, the *Acacia nigrescens* tree showed a high compensatory growth capacity, which is important for its survival [33]. In *Ziziphus mucronata*, growth was stimulated by tolerating herbivory through the production of many new shoots [34]. In other work [35], Layne and Flore showed that sour cherry trees compensate for partial defoliation by increasing the photosynthetic capacity of the remaining leaf area. In our study, increasing stomatal conductance during spring and autumn may improve water use efficiency, which implies more $CO_2$ supply for assimilation, and promotes photosynthesis [32,36]. However, the decline in photosynthetic assimilation rates in summer in both grazed and ungrazed trees suggested that drought may limit the compensation mechanism. In addition, during the summer season, $g_s$ decreased while $C_i$ exhibited elevated values in comparison with the spring season. The marked decline in $A_{Net}$ and $g_s$ in the absence of important changes in $C_i$, or increased $C_i$, suggests that the reported reduction in photosynthetic rates could be due to effects on the photosynthetic apparatus [37]. Flexas et al. [36] suggested that the increase in $C_i$ inducted by the $A_{Net}$ reduction caused stomatal closure, with a consequent decrease in $g_s$. Noteworthy, water use efficiency is given as the most excellent index for measuring carbon fixation per water consumption ratio and evaluating plant flexibility under stress conditions [38]. In our study, the WUEi values were significantly higher in grazed plants; this suggests that the trees use water more economically under grazing conditions.

The maximum photochemical efficiency of PSII (Fv/Fm) might give a simple and rapid marker to estimate the photosynthetic performance [39]. In our study, the grazed trees showed higher Fv/Fm values than ungrazed trees, indicating that grazing improved the photosynthetic performances. However, in both sites, Fv/Fm decreased in summer and recovered maximal values after the soil water increased in autumn in ungrazed trees (grazed trees could not be measured). Our results agree with data recorded in the same species in previous work [40]. Under summer stress conditions, a decrease in the Fv/Fm ratio under the optimum values (set between 0.7 and 0.8) has been widely reported, indicating photoinhibition effects [41]. Therefore, we suppose that moderate photoinhibition effects have occurred in this case because of the slight decrease of the Fv/Fm. In addition, a reduction in ΦPSII at midday was observed in summer at both sites. However, this reduction was reversible within a few minutes of dark adaptation. [42] described this behavior as photoinhibition due to photoprotective processes.

Photosynthetic pigments are potentially suitable biomarkers because they are sensitive to environmental fluctuations. Grazed trees exhibited significantly higher chlorophyll content than ungrazed trees during spring. These results suggest that grazing stimulated total chlorophyll synthesis [43] and could induce a compensation mechanism against herbivores [40]. The trees increased the photosystem components used to promote photosynthesis efficiency by improving the transfer of electrons through the reaction centers. This is in accordance with the $A_{Net}$ results. Furthermore, the significant decrease in Chla+b shared by grazed and ungrazed trees during the drought season suggests a mechanism to reduce photosynthetic system photo-damage under excessive summer radiation [44]. Trees from both sites recovered and showed maximal chlorophyll pigment content when soil water was available in autumn.

The lower Chla/b suggests that trees from both sites invested more in light energy absorption [45], as Chlb is only a light-harvesting pigment and not a light energy transforming pigment. Chla+b/Car represents a good indicator of plant stress detection and

tolerance and is sensitive to photooxidative damage. Our study indicated that grazing significantly increased Chla+b/Car, suggesting that grazed trees show a lower oxidative stress intensity [43].

Plant stomata are strongly related to plants' ecological and physiological functions, such as photosynthesis and transpiration [46]. Earlier studies detected that stomatal density is sensitive to the conditions of the abiotic environment [47], water availability being the main ecological factor affecting stomatal density [46]. Despite several works, the relationship between grazing and stomata parameters is not well known.

In our study, no clear pattern of grazing impact on studied stomatal features was observed during summer and autumn. In contrast, during the spring season, significant differences were recorded between the two sites. Grazed trees exhibit a low density of large stomata, and high photosynthetic rates. This is probably due to some modifications in the biochemical processes and the mesophyll conductance that compensate for the negative impact of stomata traits on photosynthesis [48]. The greater stomatal size promotes the diffusion of $CO_2$ into the leaf since its conductance is corresponding to the square of the effective radius of the stomatal pore, which leads to an increase in $g_s$ [49].

Grazed trees showed, in summer, a decrease in stomatal size. A small guard cell size could maintain the stomata open under dry conditions, exhibiting the balance between carbon assimilation through photosynthesis and avoidance of excessive transpiration, which is an adaptive strategy to dry conditions [49]. In addition, plants with a high density of small stomatal size can close and open faster than those with low density but large stomatal size [50,51]. Otherwise, a significant negative relationship between stomatal size and stomatal density was found in both sites (data not shown), which was consistent with previous reports [48,52,53].

## 4. Materials and Methods

### 4.1. Plant Material and Experimental Conditions

The study was performed in the Argan forest of M'sguina, located in the northeast of Agadir in the Drarga municipality (west central Morocco). At the Drarga, the forest is about 11,148 hectares, and livestock farming is extensive and consists mainly of 40,000 goats and sheep. Two experimental sites differing in grazing intensity: grazed (30°42′57.5″ N 9°40′11.7″ W) and ungrazed (30°42′67.1″ N 9°39′97.6″ W) were selected. The ungrazed site is a part of a 220 hectare piece of land. Grazing by livestock has been excluded since 1988. The grazed site is an overgrazed zone located outside the fenced area. Both sites share the same soil type and climate.

Climate is arid, with marine influence, but characterized by a long dry summer. The daily recorded climatic data of the studied areas were analyzed using data from the website NASA Power Data Access Viewer [54] (https://power.larc.nasa.gov/data-access-viewer/ accessed on 30 April 2020). In the grazed site, natural vegetation is very scarce. Both of the studied sites are dominated by the *A. spinosa* tree with the presence of some spiny shrubs, such as *Ziziphus lotus*, *Acacia gummifera*, and *Rhus tripartita*.

Ten trees of similar size were randomly selected and marked permanently in each site to measure the same individual tree throughout the study period. Field measurements were carried out when the most contrasting conditions of climate occurred, in April 2018 (favorable season), in September 2018 (water deficit and high temperature), and in early December 2018 (recovery from the stress of summer when the first autumn rains occur). At each study site, sun-exposed leaves were collected from each marked individual and immediately stored in a portable cooler. In the laboratory, samples were kept at −24 °C until use for analysis.

### 4.2. Gas Exchange Measurements

Gas exchange measurements, including $CO_2$ assimilation rate ($A_{Net}$), stomatal conductance ($g_s$), and sub-stomatal $CO_2$ concentration ($C_i$), were done by using an infrared gas analyzer (LCi-SD portable, ADC, Hertfordshire, UK). Measurements were recorded

under field conditions on fully expanded leaves (three leaves per plant for each study area). A sensor in the leaf chamber permits the reading of leaf temperature, and a sensor outside the chamber provides reading of the flux density of the incident photon of the photosynthetically active radiation (PPFD). All gas exchange measurements were conducted on cloud-free days in the morning (from 8:00 a.m. to 11:00 a.m.) during three seasons of 2018 (spring, summer, and autumn). As the leaf surface was smaller than the leaf chamber, measured leaves were scanned and the leaf surface was calculated using the software [55], then gas exchange measurements were recalculated for leaf surfaces. Instantaneous water use efficiency (WUEi) was calculated as the ratio between $A_{Net}/E$.

### 4.3. Chlorophyll Fluorescence Measurements

Chlorophyll fluorescence measurements were performed on randomly selected, fully developed leaves ($n$ = 30, three leaves per tree for each study site). Light- and dark-adapted chlorophyll fluorescence (Fv/Fm, ΦPSII) parameters were carried out at midday on the adaxial surface of the leaf in field conditions using a portable fluorimeter (OSp5+, ADC, Hertfordshire, UK) as described by [56].

After 20 min of dark adaptation, the initial minimal fluorescence of leaves ($F_0$) was measured. The maximal fluorescence in this state (Fm) was measured after applying a saturating actinic light pulse. The ratio of variable to maximal fluorescence, Fv/Fm, called maximal PSII quantum yield, was automatically calculated by the fluorimeter as:

$$\frac{F_v}{F_m} = \frac{F_m - F_0}{F_m}$$

The effective PSII quantum yield of light-adapted leaves (ΦPSII) was calculated as [57]:

$$\Phi PSII = \frac{F'_m - F}{F'_m} = \frac{\triangle F}{F'_m}$$

$F'_m$ is the maximal and F is the steady-state fluorescence under actinic irradiance (with a leaf-clip holder). The measurements were done on sun-exposed leaves.

### 4.4. Pigment Content

Pigment content was extracted with pure acetone and determined calorimetrically following [58]. The spectrophotometric absorbance of chlorophyll a, chlorophyll b, and carotenoids were determined using a VWR UV/Vis-1600PC Spectrophotometer at three wavelengths, 661.6 nm, 644.8 nm, and 470 nm. Concentrations of pigments were obtained by calculation using the Lichtenthaler equations.

### 4.5. Water Status

Relative water content (RWC) was determined following [59] methods. First, 0.5 g of fresh mass (FW) were floated in distilled water to saturation in plastic bags at 5 °C for 24 h, then weighted to obtain saturated mass (TW). The same sample was then oven-dried for 48 h at 80 °C to get dry mass (DW). RWC was calculated using the following formula:

$$RWC = \frac{FW - DW}{TW - DW} \times 100$$

### 4.6. Stomatal Features

Stomatal density (SD) was determined on three randomly selected leaves per tree for each study site during the three seasons by using the nail polish leaf impression method according to [60]. The stomata density was counted under a microscope at (×400). Stomata cell length was measured on 12 randomly selected stomata from the same impressions used for SD determinations.

*4.7. Statistical Analysis*

Analysis of variances (ANOVA) was performed using SPSS statistical software (IBM SPSS Statistics 20, Inc., Chicago, IL, USA). Two-way analysis of variance was used to test the effect of site and season in each of the plant physiological, morphological, and phytochemical study variables and leaf traits. The *t*-test was used to evaluate the significant differences in all the measured variables between the grazing-free and grazing sites. The $p < 0.05$ values were considered the statistical tests' significant level.

**5. Conclusions**

In conclusion, the photosynthetic response of the studied trees varied between grazing conditions and through seasons. Our results showed that grazing has a positive impact on the photosynthetic performances of *A. spinosa* during the favorable season. Trees under grazing conditions increased photosynthetic activity in leaves to compensate for the loss of leaf area after herbivore damage. This enhancement of photosynthetic rate may be explained by increased stomatal conductance. In addition, they use water more economically and show lower oxidative stress. This finding confirms that Argania is a tolerant species against herbivores. Nevertheless, summer drought limits the compensation mechanism and negatively affects the physiology of the tree.

Since the key to preserving the future of forests is sustainable forest management, we suggest that proper grazing management can be a control tool to increase plant performance and improve species resilience. In addition, a study including different defoliation intensities will reveal the tolerance threshold of Argania towards herbivores.

**Author Contributions:** Conceptualization, A.N.D., F.A.-L. and A.A.A.-N.; methodology, A.N.D., L.B. (Laila Boukhalef), F.A.-L., A.E.A., L.B. (Laila Bouqbis) and M.T.; software, A.N.D.; validation, F.A.-L.,A.E.A. and A.A.A.-N.; formal analysis, A.N.D., L.B. (Laila Boukhalef), A.E.A. and M.T.; investigation, A.N.D. and L.B. (Laila Boukhalef); resources, A.N.D. and L.B. (Laila Boukhalef); data curation, A.N.D., L.B. (Laila Boukhalef) and F.A.-L.; writing—original draft preparation, A.N.D. and F.A.-L.; writing—review and editing, A.N.D., F.A.-L., A.A.A.-N., K.E.M. and L.B. (Laila Bouqbis); supervision, F.A.-L., A.E.A. and K.E.M.; project administration, F.A.-L.; funding acquisition, F.A.-L. and A.A.A.-N. All authors have read and agreed to the published version of the manuscript.

**Funding:** This work has been supported by Ibn Zohr University "Projets de l'Université Ibn Zohr 2016–2018", and King Abdulaziz City for Science and Technology. A.N.D. and L.B. (Laila Boukhalef) were supported by a grant from Ministère de l'Enseignement Supérieur.

**Institutional Review Board Statement:** Not applicable.

**Informed Consent Statement:** Not applicable.

**Data Availability Statement:** Data are contained within the article.

**Acknowledgments:** We thank Laknifli A., the Polydisciplinary Faculty of Taroudant and Sinan F., the Faculty of Applied Sciences—Ibn Zohr University, for the facilities they generously provided for this scientific research. The valuable comments of anonymous reviewers are acknowledged.

**Conflicts of Interest:** The authors declare no conflict of interest.

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
