# Peer review of "Photosynthetic Behavior of Argania spinosa (L.) Skeels Induced under Grazed and Ungrazed Conditions"

_sustainability, doi:10.3390/su141912081_

Round 1

Reviewer 1 Report

A study of the ecophysiological performance of forest-forming plant species under stressful conditions remain relevant. The title of the article corresponds to its content. The authors formulated a goal clearly and used an appropriate research design and methods. The results are well presented. The conclusions are supported by the results. However, there are some remarks.

1.      In Materials and Methods in 4.2. Gas exchange measurements it is necessary to describe the conditions for measuring parameters - illumination, CO2 ppm, temperature, and air humidity - since the rate of photosynthesis depends on the level of these factors. This information should be used when discussing the results.

2.       In Discussion remove or correct the sentence (lines 178-179) because cause and effect are reversed. Change in the stomata conductivity is the cause of the WUE change and not vice versa.

3.       In Discussion (line 191) remove incorrect citation. In [32] tree WUE have not been studied.

4.       Authors compared their results only with the results of studies that were carried out on grasses and forbs under grazing. It is necessary to note this in the discussion, since different life forms react differently to the stress effect, and also to find and use articles on the reaction of trees to grazing in the discussion.

5.       Since the conclusion should be based on own results, it is necessary to remove the citation of the results from other studies (lines 331-332).

Author Response

Dear reviewer,

I would like to thank you for your valuable corrections. You will find below my reply to your relevant remarks.

  1. In Materials and Methods in 4.2. Gas exchange measurements it is necessary to describe the conditions for measuring parameters - illumination, CO2 ppm, temperature, and air humidity - since the rate of photosynthesis depends on the level of these factors. This information should be used when discussing the results.

Response 1:

The LCPro-SD (ADC, UK) Infrared gas exchange analyzer permits a fully programmable environmental control. However, unfortunately, our gas exchange analyzer, the LCI-SD, does not permit programmable environmental control. Therefore, gas exchange parameters were recorded under field conditions, which varied through seasons. In addition, the sensor in the leaf chamber permits the reading of leaf temperature, and the sensor outside the chamber provides readings of the flux density of the incident photon of the photosynthetically active radiation (PPFD). We have corrected and specified in materials and methods that it is the LCi-SD type.

  1. In Discussion remove or correct the sentence (lines 178-179) because cause and effect are reversed. Change in the stomata conductivity is the cause of the WUE change and not vice versa.

Response 2:

Modified as recommended.

  1. In Discussion (line 191) remove incorrect citation. In [32] tree WUE have not been studied.

Response 3:

Modified as recommended.

  1. Authors compared their results only with the results of studies that were carried out on grasses and forbs under grazing. It is necessary to note this in the discussion, since different life forms react differently to the stress effect, and also to find and use articles on the reaction of trees to grazing in the discussion.

Response 4:

Modified as recommended.

  1. Since the conclusion should be based on own results, it is necessary to remove the citation of the results from other studies (lines 331-332).

Response 5:

Modified as recommended.

Reviewer 2 Report

Comments to the Authors

The authors have presented good work in the manuscript. The manuscript seems so complete to me that I don’t see any shortcoming in the paper. Thus the paper can be accepted in its current form.

Author Response

Dear reviewer,

I would like to thank you for giving your time to read our paper.
Kind regards.